# Natural Extracts Mitigate the Deleterious Effects of Prolonged Intense Physical Exercise on the Cardiovascular and Muscular Systems

**DOI:** 10.3390/antiox12071474

**Published:** 2023-07-22

**Authors:** Marc Yehya, Doria Boulghobra, Pierre-Edouard Grillet, Pablo R. Fleitas-Paniagua, Patrice Bideaux, Sandrine Gayrard, Pierre Sicard, Jérome Thireau, Cyril Reboul, Olivier Cazorla

**Affiliations:** 1PhyMedExp, INSERM, CNRS, CHU Montpellier, University of Montpellier, 34295 Montpellier, France; marc.yehya@univ-avignon.fr (M.Y.); pe-grillet@chu-montpellier.fr (P.-E.G.); pablo.fleitaspaniagu@ucalgary.ca (P.R.F.-P.); pierre.sicard@inserm.fr (P.S.); jerome.thireau@inserm.fr (J.T.); 2UPR-4278, Laboratoire de Physiologie Expérimentale Cardiovasculaire, Avignon University, 84029 Avignon, Francesandrine.gayrard@univ-avignon.fr (S.G.); cyril.reboul@univ-avignon.fr (C.R.); 3Département de Biochimie et d’Hormonologie, CHU Montpellier, 34295 Montpellier, France

**Keywords:** fatigue exercise, muscle, heart, natural extracts

## Abstract

Muscle fatigue is a common symptom induced by exercise. A reversible loss of muscle force is observed with variable rates of recovery depending on the causes or underlying mechanisms. It can not only affect locomotion muscles, but can also affect the heart, in particular after intense prolonged exercise such as marathons and ultra-triathlons. The goal of our study was to explore the effect of four different natural extracts with recognized antioxidant properties on the contractile function of skeletal (locomotion) and cardiac muscles after a prolonged exhausting exercise. Male Wistar rats performed a bout of exhausting exercise on a treadmill for about 2.5 h and were compared to sedentary animals. Some rats received oral treatment of a natural extract (rosemary, buckwheat, Powergrape^®^, or rapeseed) or the placebo 24 h and 1 h before exercise. Experiments were performed 30 min after the race and after 7 days of recovery. All natural extracts had protective effects both in cardiac and skeletal muscles. The extent of protection was different depending on muscle type and the duration post-exercise (just after and after one-week recovery), including antiarrhythmic effect and anti-diastolic dysfunction for the heart, and faster recovery of contractility for the skeletal muscles. Moreover, the muscular protective effect varied between natural extracts. Our study shows that an acute antioxidant supplementation can protect against acute abnormal endogenous ROS toxicity, induced here by prolonged exhausting exercise.

## 1. Introduction

Regular physical exercise is crucial for maintaining good health, especially in reducing the risk of mortality from all causes, as well as the risk of cardiovascular disease, cancer, and diabetes [1]. Since the legendary death of Philippides just after running from Marathon to Athens, the potential deleterious effects of acute prolonged exhausting exercise (PEE) on health have been debated. With the growing popularity of marathons, triathlons, and other physically demanding events, PEE could become a public health issue.

Muscle fatigue is a frequent symptom caused by intense and prolonged physical activity. During or shortly after exercise, muscle fatigue results in a temporary decrease in muscle force (muscle contractility). It can manifest either shortly after starting exercise (acute muscle fatigue) or after a prolonged period of high-intensity exercise (delayed exercise-induced fatigue). The latter is characterized by tiredness occurring only after a prolonged duration of continuous exercise. Muscle fatigue can vary in terms of underlying mechanisms, causes, and rates of recovery. 

Oxidative stress plays a significant role in muscle fatigue, particularly during prolonged and intense exercise. Muscle contractions generate free radicals, leading to oxidative stress and damage to cellular components [2]. Reactive oxygen species (ROS) levels increase during physical exercise, contributing to both muscle fatigue and the recovery process. The restoration of muscle force after fatiguing exercise is a slow process that may take several days to complete. ROS are believed to be crucial in the delayed recovery observed in such cases [3]. Intense muscle contractions during fatigue cause an elevation of superoxide (O_2_^•−^) and its byproducts, including hydrogen peroxide (H_2_O_2_), hydroxyl radical, or peroxynitrite radical in various muscle compartments. Mitochondrial electron transport chain and NADPH oxidase (NOX) complex are the primary sources of O_2_^•−^ in muscles [2]. ROS can exert various effects on different aspects of muscle contraction and relaxation, encompassing energy levels, excitation, calcium homeostasis, and properties of the contractile proteins [3]. Direct exposure of skeletal muscle to ROS induces functional changes that resemble the characteristics of muscle fatigue experienced during exercise. For instance, Andrade et al. [4] demonstrated that exposing intact muscle fibers of mice to hydrogen peroxide leads to a decline in specific force during submaximal tetanic contractions.

The heart can also exhibit signs of reversible dysfunction induced by exercise. Dilation of the right atrium and ventricle, along with elevated cardiac troponin and natriuretic peptide levels, have been reported in about one-third of marathon runners [5,6]. Prolonged exhaustive exercise is also known to cause transient dysfunction of the left ventricle (LV) associated with myocardial damages [7,8]. A temporary reduction in LV diastolic relaxation during early recovery is typical, without necessarily changes in systolic function [8]. This condition is referred as exercise-induced cardiac fatigue. Our group has demonstrated that under extreme cardiac stress induced by PEE, adrenergic and oxidative signaling pathways work in concert to modify myofilament properties that alter cardiac function [9]. We have also shown that intraperitoneal injection of a general antioxidant treatment can prevent some of the molecular and tissue dysfunctions. Whether it is helpful to apply exogenous antioxidant supplements under such conditions to prevent muscle fatigue is still under debate.

The goal of our study was to test whether acute treatment with natural extracts containing antioxidant properties before a prolonged exhausting exercise bout can (1) reduce or prevent damages in the skeletal and cardiac muscles, or (2) improve their recovery. We selected four natural extracts based on their recognized antioxidant properties, including rosemary (*Rosmarinus officinalis*, Lamiaceae) [10], buckwheat (*Fagopyrum esculentum*) [11], Powergrape^®^ [12], and rapeseed (*Brassica napus* L.) [13,14]. The study did not aim to comprehensively analyze each antioxidant compound in terms of bioavailability and antioxidant mechanisms, but rather to assess the ability of well-characterized natural antioxidant compounds to protect muscles during intense exercise. The study investigated the contractility of isolated skeletal muscles (fast-Extensor Digitorum Longus, EDL, and slow-soleus) ex vivo, examined cardiovascular function in isolated aortic rings and in isolated perfused hearts ex vivo, and examined it in vivo using echocardiography and electrocardiography. Our study reveals that acute antioxidant supplementation can protect against acute abnormal endogenous ROS toxicity induced by prolonged exhausting exercise. The extracts exhibited varying beneficial effects depending on the muscle type and the duration after exercise including antiarrhythmic effect and anti-diastolic dysfunction for the heart, and faster recovery of contractility for the skeletal muscles.

## 2. Materials and Methods

*Animal studies*. Male Wistar rats (12 weeks old, *n* = 108; weight 308 ± 4 g; Janvier Laboratories, Le-Genest-Saint-Isle, France) were housed on a 12 h light–dark cycle and had free access to water and food. All investigations conformed to European Parliament Directive 2010/63/EU and were approved by the local ethics committee rules Comité d’éthique pour l’expérimentation animale Languedoc-Roussillon (N° CEEA-00322.03). The rats were euthanized with a sodium pentobarbital overdose (140 mg/kg, i.p.).

The rats were familiarized with the treadmill by running at 15 m/min without slope (about 65% of the maximal aerobic velocity) for 15 min daily for a week before the experiment. Some rats then underwent a prolonged exhausting exercise (PEE) on a treadmill. The exercise was performed at 1% inclination with incremental running speed, starting at 13 m/min and increasing every 15 min to 17 m/min for at least 2 h. Then, the speed was progressively increased every 15 min by 2 m/min until exhaustion (166 ± 3 min, Appendix A), as stated previously [9]. Prior to the exercise, the rats were fed by intragastric gavage 24 h and 1 h before PEE with either a placebo or one of four natural extracts with antioxidant properties: rosemary, buckwheat, Powergrape^®^, or rapeseed. The doses of the extracts were consistent with human daily doses: rosemary blend GAX00067 (87.4 mg/kg body weight), buckwheat (76.8 mg/kg body weight), Powergrape^®^ GAX00069 [12] (76.8 mg/kg body weight), and rapeseed (105.6 mg/kg body weight). The extracts, provided by Givaudan-Naturex (Avignon, France), were diluted in carboxymethylcellulose sodium salt 0.5% (*w*/*v*) (CMC; Ref# C4888, Batch number: SLBR1692V, SIGMA ALDRICH, St Quentin Fallavier, France) and dissolved in distilled water at room temperature. The placebo received only CMC. Animals were regularly given access to water during the exercise protocol to avoid dehydration and hypovolemia. The experiments were performed 30 min after PEE (Exer) and after 7 days of recovery (Recov). The animals that exercised were compared with non-runner rats (control group), except for ECG and echocardiography, where the animals were their own controls before the exercise. All investigators were blinded to the treatment groups.

### 2.1. Transthoracic Echocardiography

A high-resolution ultrasound was performed three days before, 30 min, and 7 days after running using a Vevo 3100 (Fujifilm Visualsonics, Tokyo, Japan), equipped with a 21 MHz transducer (MX250), as shown previously [15]. Rats (*n* = 10/group) were anesthetized with 2% isoflurane. Left ventricular (LV) parasternal long axis 2D view in M-mode was performed at the level of papillary muscle to assess LV wall thicknesses and internal diameters, allowing the calculation of the fractional shortening (FS) and ejection fraction (EF) by the Teicholz method. The relative wall thickness index (RWT) was calculated as RWT = (IVSd + PWTd)/LVIDd (with IVS: inter-ventricular septum; PWT: posterior wall thickness; LVID: LV internal diameter; d, in diastole). EF was also calculated (%) from a B-mode parasternal long axis view by tracing endocardial end-diastolic and end-systolic borders to estimate LV volumes.

Mitral inflow was recorded by a pulsed-wave Doppler in the apical four-chamber view by placing the sample at the tip of the mitral valves. Peak early (E) and late atrial contraction (A) mitral inflow waves were measured, and the E/A ratio was calculated. Tissular doppler imaging of the mitral annulus was performed to assess the early and late diastolic myocardial relaxation velocity waves (e′ and a′). All measurements were quantified and averaged for three cardiac cycles using the offline vevolab 3.5.1 software.

### 2.2. Electrocardiogram Recordings and Analyses

Rats (*n* = 8/group) were equipped with telemetric transmitters (CA-F40, Data Sciences International, St. Paul, MN, USA) inside the peritoneal space under general anesthesia (isoflurane 2.5% in O_2_), as shown previously [16]. After 14 days of housing for total recovery and acclimatization, two-lead ECG (DII) were recorded in baseline, just after, and 7 days after the intensive exercise using a signal transmitter-receiver (RPC-1, Data Sciences International, St. Paul, MN, USA) connected to a data acquisition system (Ponemah Physiology Platform, DSI) at a sampling rate of 2 kHz. Heart rate as the RR interval, as well as the QT interval in milliseconds, were measured. The QT interval was defined as the time between the first deviation from an isoelectric PR interval until the return of the ventricular repolarization to the isoelectric TP baseline from lead II ECGs. Since recordings were performed in awake rodents, including rats that do not have typical rate-adaptation of their QT interval, QT was not corrected to heart rate [17]. Ventricular arrhythmia as ventricular extrasystole was identified based on abnormal RR interval, verified, and counted by hand in a blinded condition over the entire ECG period. Arrhythmia was given as a number of events/30 min ECG period.

### 2.3. Isolated Perfused Hearts

Cardiac function was studied ex vivo in an isolated perfused heart, as previously described [9]. After anesthesia (100 mg/kg sodium pentobarbital, i.p.) and total loss of consciousness, rats were heparinized (1000 UI/kg, i.v.). The heart was rapidly removed and immersed in ice-cold Krebs solution. The aorta was cannulated for perfusion with oxygenated (95% O_2_/5% CO_2_) Krebs solution (118.3 mmol/L NaCl, 25 mmol/L NaHCO_3_, 4.7 mmol/L KCl, 1.2 mmol/L MgSO_4_, 1.2 mmol/L KH_2_PO_4_, 11.1 mmol/L glucose, 1.25 mmol/L CaCl_2_, pH = 7.4, 37 °C). The right atrium was excised. The atrioventricular node was crushed using fine forceps and the heart was paced at a rate of 300 beats/min with an electrical stimulator (Low voltage stimulator, BSL MP35 SS58L, 3V). A non-compliant balloon was inserted into the LV via the mitral valve and the balloon volume was adjusted to achieve an LV end diastolic pressure of 5 mmHg. The heart was perfused at a constant pressure (80 mmHg) and was stabilized for 30 min. Throughout the procedure, the cardiac function parameters were recorded (MP35, BioPac System Inc., Goleta, CA, USA) to calculate the developed pressure (DevP), as well as the maximal (dP/dt_max_) and minimal (dP/dt_min_) first derivatives of LV pressure.

### 2.4. Measurement of Soleus and EDL Contractile Properties

The entire soleus or EDL was isolated from the hindlimbs and mounted between a lever arm of a position feedback servomotor (model 6650LR, Aurora Scientific, Aurora, ON, Canada) to measure isometric contractile properties at 28 °C, as previously described [18]. The muscle was stretched at L0 (the length at which the muscle produced maximal isometric tension) and then was supramaximally stimulated using square wave pulses (Model S48; Grass Instruments, West Warwick, RI, USA). The force–frequency relationship was determined by sequential stimulation of the muscles for 600 ms at 10, 20, 30, 50, 60, 80, 100, and 120 Hz with 1 min intervals between each stimulation train. Muscle fatigue was assessed by measuring the rate of muscle force loss during repetitive 2 ms-pulse stimulation every second at 30 Hz for 300 ms, over a 5 min period. The muscle cross-sectional area was determined by dividing the muscle’s weight by their length at L0 and tissue density (1.056 g/cm^3^). EDL or Soleus force production was then normalized to the muscle cross-sectional area to determine their specific force, expressed in newton per square centimeter (N·cm^−2^).

### 2.5. Vascular Reactivity on Isolated Aortic Rings

Isolated aortic rings procedure was performed, as previously described [19]. Briefly, aortic rings were mounted on stainless steel connected to an isometric force transducer (EMKA Technologies, EMKA Paris, France). The resting tension was adjusted to 2 g corresponding to the optimal length for tension development in the aorta of 12-wk-old rats. The rings were then equilibrated for 60 min. After equilibration, test doses of phenylephrine (PE) and acetylcholine (ACh) were added to the rings to ensure reproducibility of constriction, relaxation, and endothelial integrity. After being rinsed, each vessel ring was precontracted with PE (10^−6^ M). After the pre-constriction reached a plateau, endothelium-dependent relaxation was obtained by cumulative ACh concentrations (10^−9^ to 10^−4^ M), and endothelium-independent relaxation was obtained by cumulative Sodium Nitroprusside (SNP; 10^−9^ to 10^−4^ M) concentrations.

### 2.6. Statistical Analysis

Data were presented using mean ± SEM values. The power and sample size have been determined using G*power software 3.1. Statistics were performed using GraphPad Prism software 9.3.1 (GraphPad Software, La-Jolla, San Diego, CA, USA). Data were subjected to Student *t*-test, one-way ANOVA, repeated measures ANOVA, or two-way ANOVA test. When normality failed, the non-parametric Friedman ANOVA was used. When significant interactions were found, a Bonferroni post hoc test was applied with *p* < 0.05.

## 3. Results

### 3.1. PEE Induced a Loss of Isometric Strength in EDL and Soleus Muscles

All animals ran on the treadmill for about 2.5 h until exhaustion (Appendix A). The study evaluated the muscle-specific force–frequency relationships of slow-twitch soleus and fast-twitch EDL muscles in sedentary rats and 30 min (Exer) or one-week (Recov) after a prolonged exhausting exercise (PEE) (Figure 1A). The study found that the force produced by the soleus muscle decreased for most stimulation frequencies in the placebo group compared to the sedentary group, with a loss in tetanus force of about 63% (Figure 1B–D). One week later, the recovery of force production was incomplete in the soleus muscle (17% lack of recovery of the tetanic force, *p* = 0.1) compared to sedentary animals (Figure 1B). The EDL muscle was weakly affected by PEE with a normal twitch force and a decrease in the tetanic force by about 30%, which did not reach significance (*p* = 0.07) (Appendix A). The recovery of force was complete after a week.

The animals were orally fed with different natural extracts (rosemary, buckwheat, Powergrape, and rapeseed) 24 h and 1 h before exercise and ran the same duration until exhaustion as the placebo rats (Appendix A). For the soleus muscle, the natural extracts had no effect on the isometric force decline post-exercise (Figure 1C,D). One week later, we observed a complete recovery of tetanic force in rosemary and rapeseed groups (Figure 1D). For the EDL muscle, the natural extracts had no effect on the isometric force post-exercise (Appendix A). After the recovery period, the force developed was higher with buckwheat, and tended to be higher with rosemary and rapeseed (*p* = 0.07 and 0.12, respectively) when compared with post-exercise values.

Taking all these results into account, we concluded that the treadmill PEE caused a loss of muscle strength generation particularly in the slow-twitch soleus, just after exercise. One week later, the recovery was incomplete only in the soleus. Natural extracts improved the recovery of force development in the soleus muscle, in particular rosemary, Powergrape, and rapeseed extract, compared to the placebo group. Natural extracts had minor effects on EDL muscle.

### 3.2. PEE Induces Significant Fatigue in Skeletal Muscles

Considering the long exercise duration performed by the animals (157.5 ± 6.2 min), we also evaluated their level of muscle fatigue and ability to maintain repetitive sub-maximal contractions (Figure 2A,B). We observed that PEE decreased the endurance strength by 80% of soleus muscle in the Placebo post-exercise group compared to the control sedentary group (Figure 2A,B). Even after a week of recovery, the endurance capacity remained incomplete in soleus muscles (Figure 2A,B). The EDL muscles were less affected, with a loss of endurance strength by 30–40% and incomplete recovery after a week (Appendix A)

The natural extracts had no major effect on endurance strength post-exercise compared with the placebo group in soleus (Figure 2C). However, after a week of recovery, all extracts enhanced the recovery of soleus-muscle endurance, as shown by the higher strength in the four treated groups compared to the placebo group (Figure 2C,D), as well as better endurance, as indexed by the higher T_1/2_ fatigue in the treated groups compared to the placebo group (Figure 2E). The natural extracts had no major effect on endurance strength of the EDL muscles post-exercise and no impact after a week of recovery (Appendix A).

Taking all these results into consideration, we concluded that the treadmill PEE caused a loss of muscle endurance of the skeletal muscles that recovered incompletely after a week in both muscle types. The natural extracts improved the recovery of endurance performances, in particular in the slow-twitch soleus.

### 3.3. PEE Does Not Alter Vascular Function in Isolated Aorta

The vascular function was assessed by examining isolated aortic rings. The maximal contraction induced by phenylephrine was not affected by the exhausting exercise (Ctrl: 2.84 ± 0.69 g; Placebo: 2.37 ± 0.68 g; *t*-test, *p* > 0.05). After pre-constriction with PE, we evaluated the relaxation properties of the vessel induced by Ach or the NO-donor SNP. Exhausting exercise tended to increase the endothelium-dependent response to Ach in the placebo animals, as evidenced by the leftward shift of the curve and higher Rmax (*p* = 0.08). The endothelium-independent response to SNP was unaffected, suggesting that the exhausting exercise affected the endothelial function but not the vascular smooth muscle layer properties (Figure 3D). The different natural extracts had no effect on both the endothelium-dependent (Figure 3D) and endothelium-independent (Figure 3B) vasodilator responses.

### 3.4. Effects of Natural Extracts on the PEE-Induced Cardiac Relaxation Defaults

The cardiac function was evaluated in vivo using echocardiography on the same animal at baseline (base), 30 min after exercise, and after a week of recovery. No differences in heart rate during the exams were observed between the groups. In the placebo group, PEE did not affect the systolic function, as indicated by the unchanged ejection fraction (EF, Figure 4B). Natural extracts had no immediate effect after exercise. Interestingly, the rosemary and Powergrape^®^ groups had higher systolic contractile function by about 14% after a week of recovery compared to after exercise. Relaxation, an important property of the heart, was altered after exercise, as indicated by the decrease in indexes of diastolic function such as the E/A ratios of early diastolic and atrial transmitral inflow velocities (Figure 4C) as well as the e′/a′ ratios obtained by tissue Doppler imaging (Figure 4D). Recovery of those indexes even 1 week after exercise was incomplete in placebo-treated animals. Rosemary and Powergrape^®^ prevented the diastolic dysfunctions induced by exercise, while buckwheat and rapeseed had more modest beneficial effects.

To evaluate whether the results obtained in vivo by echocardiography were due to a direct impact on the cardiac muscle, we also examined the potential of natural extracts to protect the heart following PEE ex vivo in the Langendorff perfused heart isolated from either sedentary rats (Ctrl) or rats that underwent PEE 30 min prior. This preparation is ideal for studying the intrinsic properties of cardiac function by controlling the heart rate, loading conditions, and perfusion media [20]. Consistent with previous work [9,21], systolic function was altered after exhausting exercise, as indicated by lower LV developed pressure (Figure 5C) and contraction velocity dP/dtmax (Figure 5D) compared to sedentary hearts. LV diastolic function, indexed by relaxation velocity dP/dtmin, was also decreased following exhausting exercise when compared to sedentary animals (Figure 5E, *p* < 0.05 Control vs. Placebo, *t*-test). Hearts from animals pre-treated with natural extracts were protected against most of the contractile alterations when compared to sedentary hearts (Figure 5).

### 3.5. Effects of Intense Exercise and Natural Extracts on Cardiac Electrical Activity

The heart’s electrical activity was monitored using telemetry ECG on the same animals before and after exercise (Figure 6). As expected, the intense exercise had an impact on electrical activity. The RR interval (which is the inverse of the heart rate) was significantly reduced by PEE compared to before exercise (Base), indicating a higher heart rate or tachycardia effect. This increase in frequency is linked to the prevalence of activation of the sympathetic system during running. However, the RR interval returned to normal levels after the recovery phase (Figure 6B).

The sinoatrial (PR) and ventricular (QRS) conduction times were minimally affected by exercise (Appendix A). In contrast, the ventricular depolarization and repolarization time (QT duration) increased after running and remained high even after the recovery phase (Figure 6C).

Prolongation of the QT duration is also considered a risk factor for triggering ventricular arrhythmias [22]. The total number of arrhythmic events increased after exercise and normalized after the recovery phase (Figure 6E). The arrhythmias observed immediately after the exercise phase were mainly ventricular premature beats. Premature ventricular contractions, trigeminy and bigeminy complexes, and doublets were noted. The animals also presented bundle branch block-type conduction disturbances and second-degree atrioventricular blocks.

The natural extracts had no impact on RR interval immediately after exercise. After 1-week recovery, buckwheat, Powergrape^®^, and rapeseed decreased the heart rate, as suggested by the higher RR interval when compared to the baseline (Figure 6B). None of the treatments could prevent the increase in QT duration after exercise. However, all treatments favored the QT duration recovery after a week, particularly with rapeseed (Figure 6C). Another notable difference between the placebo group and the treated groups was the occurrence of ventricular arrhythmias just after exercise that disappeared, except in the buckwheat-treated group.

## 4. Discussion

The use of antioxidants in sports is a subject of controversy due to conflicting evidence on their effects on athletic performance. Endogenously produced ROS play crucial roles as signaling molecules that regulate cellular functions such as homeostasis, differentiation, proliferation, repair, and aging. As a result, popular antioxidant supplements like NAC, GSH, and vitamins E and C have shown little to no significant health benefits in most long-term, well-controlled clinical trials. However, our study suggests that acute antioxidant supplementation can protect against acute abnormal endogenous ROS toxicity induced by prolonged exhausting exercise. Our findings reveal protective effects in both cardiac and skeletal muscles, although to varying degrees depending on the duration after exercise (i.e., 30 min post-exercise and one week recovery). Additionally, the degree of muscular protection differed among the natural extracts used in the study (Table 1).

To evaluate the anti-fatigue effects of the natural extracts, we subjected the animals to forced running for at least 2 h until exhaustion, as conducted previously [9,21]. In this rat model, we observed the production of oxidative stress, resulting in cardiac dysfunction and damage, which could be prevented by acute antioxidant pre-treatments. Shindoh et al. [23] were the first to provide evidence suggesting that oxidants generated during exercise contribute to peripheral fatigue. They achieved this by experimentally inhibiting muscle fatigue in situ of the diaphragm in anesthetized rabbits using N-acetylcysteine, a cysteine prodrug that replenishes the intracellular GSH levels and antioxidant compound. However, the effects of ROS are highly complex and some studies suggest that reducing ROS levels during fatigue could impede ROS-mediated signaling; thus, performance would be compromised. For example, the application of H_2_O_2_ temporarily enhances the myofibrillar Ca^2+^ sensitivity [3,4], and certain experiments indicate an increase rather than a decrease in myofibrillar Ca^2+^ sensitivity after fatiguing contractions [24]. Most studies investigating antioxidant supplements consumed by endurance athletes have focused on skeletal muscles, with the aim of reducing exercise-induced oxidative stress, improving recovery, and enhancing performance [25].

The natural extracts used in this study have different compositions and antioxidant properties, resulting in different bioactive effects. Rosemary extract (*Rosmarinus officinalis*, Lamiaceae) has been approved by the European Union (EU) for food preservation. This extract contains various polyphenolic compounds, including carnosic acid, carnosol, rosemanol, rosmarinic acid, and others [26]. The presence of anti-inflammatory and antioxidant properties in rosemary extract justifies its potential protective effect [27]. Buckwheat (*Fagopyrum esculentum*), known for its rich bioactive compound content such as flavonoids and fagopyrin, is particularly abundant in rutin and quercetin, both of which are recognized as potent antioxidants [28]. Rutin, quercetin, and other flavonoids contribute to the functional and medicinal properties of buckwheat, making it increasingly popular. Purple grapes and their derivatives possess high antioxidant, anti-inflammatory and antihypertensive properties due to their rich composition in flavonoids, including various flavanols and anthocyanins [29]. In this study, a commercially approved grape extract (previously used in a clinical study with athletes) was chosen. In this previous work, supplementation with Powergrape^®^ improved the athletes’ antioxidant status, as measured by plasmatic ORAC, reduced the plasmatic exercise-induced oxidative stress, minimized the cell damage in the muscles and plasma during exercise, and improved performance and recovery capacity [12]. Finally, rapeseed (*Brassica napus* L.) is rich in phenols and flavonoids, providing it with strong antioxidant potential [30]. Recently, we discovered that sinapine, the predominant phenolic sinapic acid derivative found in rapeseed, effectively enters the mitochondria, selectively reduces the level of mitochondrial oxidative stress, and efficiently limits ROS production during cardiac ischemia reperfusion [14].

The literature suggests that skeletal muscles are affected by prolonged exhausting exercise, with the fast-twitch fibers being more susceptible to fatigue and damage than the slow-twitch fibers [2]. Here, both muscle types exhibited fatigue and delayed recovery, but to a larger extent in the soleus muscle, which was more sensitive to antioxidant treatment. The difference between muscle types may be due to the degree of solicitation during exercise, which also plays a role in muscle susceptibility. Another explanation may be the difference of the NOX expression and the ROS detoxification site between muscle types. Alves et al. [31] showed that twenty-four hours after one strenuous session of exercise, glycogen content was more reduced and NOx expression was increased in slow fibers compared to fast fibers. They also showed that mitochondria are the fundamental site in ROS detoxification in slow fibers and at a cytoplasmic level in fast fibers. During intense exercise, oxidative stress increases due to the generation of free radicals, which can lead to muscle damage and fatigue [32]. Low levels of oxidants during a resting period or in early stages of fatigue may actually help to delay the onset of fatigue and strengthen force production [32,33]. However, prolonged and intense exercise can result in elevated ROS levels, which can impair muscle function and delay recovery [2]. Recovering full muscle strength production after strenuous exercise is a slow process that can take several days [32]. While natural extracts did not prevent fatigue in our study, they did improve recovery. Antioxidants have been shown to delay exercise-induced muscle fatigue and improve exercise performance in animal and human studies [23,34,35,36,37,38,39,40]. However, the effects of antioxidants are complex and may vary depending on the type and duration of exercise. For example, N-Acetyl cysteine (NAC) has been shown to attenuate low-frequency force decline during fatigue, but does not improve the recovery after fatigue [23,35,41]. Moreover, antioxidant supplements such as vitamins C and E, as well as carotenoids, may interfere with muscle adaptations that require oxidative stress during chronic training and should not be recommended.

Our study confirmed that prolonged intense exercise impacts several muscular functions, but to a different extent depending on the muscle investigated. Cardiac function is altered both in vivo and ex vivo, particularly the relaxation capacities of the heart, which confirms our previous results [9,21]. In this study, we further revealed that PEE could alter the electrical activity of the heart, as indexed by the QT prolongation. This increased QT duration without prolongation of QRS suggests an increase in ventricular repolarization, which means the heart muscle takes longer than normal to recharge between beats. More importantly, excessive QT prolongation can trigger tachycardia and arrhythmias, which was a novel feature of our model that was observed just after exercise. After a week of recovery, some indexes remained altered, suggesting intrinsic profound cellular modifications. Natural substances could have pro-arrhythmic effects because they contain bioactive molecules. The hERG channel blockage by small molecules are associated with increased risk of fatal arrhythmias. The hERG channel blocking profile of natural compounds present in frequently consumed botanicals (i.e., dietary supplements, spices, and herbal medicinal products) is not routinely assessed. In the literature, most studies investigated hERG channel blockers from natural sources as antiarrhythmic but not as pro- arrhythmic. Herbal medications are commonly used for clinical purposes, including the treatment of cardiovascular conditions. Potentially relevant side effects, including increased risk of drug interactions, are described [42]. In our study, we observed antiarrhythmic effects of the natural extracts. Based on our experience and previous publications, if the compounds we used were pro-arrhythmic, we would have seen a large increase in arrhythmias after this kind of cardiac stress that mix adrenergic stress, oxidative stress, and inflammation stress.

The vascular function was not affected ex vivo by PEE. However, the abdominal aorta is the largest blood vessel in the body, which is very compliant and aims to support and propagate pulsatile ventricular output. Although acute PEE does not affect compliant arteries, it cannot be excluded that resistance arteries, i.e., small arteries and arterioles, may be affected. Further studies will be necessary to evaluate the impact of PEE on the vascular bed.

Nevertheless, as previously reported by our group [9,21] and others [43], overproduction of ROS during acute prolonged exhaustive exercise can trigger muscle dysfunctions and have deleterious health impacts [44]. Thus, in the specific context of professional or amateur athletes who regularly perform such exhausting exercise, it is interesting to identify protective strategies to prevent pathological remodeling and improve cardiac and muscle recovery. Our results indicate that rosemary and Powergrape^®^ were good natural extracts for the heart post-exercise (anti-arrhythmic and preserve relaxation), and rapeseed during recovery. The same three extracts also had beneficial effects on skeletal muscle functions during recovery. Those compounds have both anti-inflammatory and antioxidant properties that may mediate the protective effect. Previous studies have shown that targeting excessive ROS production with NAC [9] or Apocynin [21] prevents cardiac dysfunctions. The same results were reported for skeletal muscles with the use of Pyrroloquinoline quinone (PQQ) [43], anthocyanins [45], astaxanthin [46], Catechins of green tea [47], and Curcumin [48]. Thus, despite the controversial nature of using antioxidants during exercise training, we report that acute pre-treatments with natural plant extracts recognized for their antioxidant properties could be interesting to limit both skeletal and cardiac muscle damages. This is consistent with previous studies showing that polyphenols derived from pomegranates, cherries, and blueberries reduced muscle soreness and improved muscle strength after eccentric exercise [49,50,51]. This strategy with natural extracts could be useful in preventing the potential health impacts of such damages and in promoting functional recovery. In our rat model, this is particularly true for rosemary, grape, and rapeseed natural extracts. Polyphenols, including flavonoids derived primarily from fruits, have been of interest due to their antioxidant and anti-inflammatory effects [52]. Thus, it cannot be excluded that some of our beneficial effects are due to anti-inflammatory action. It is important to recognize that different antioxidant frontlines are activated during physical exertion and in the subsequent recovery period. The dynamic nature of antioxidant responses underscores the complex interplay between oxidative stress and the body’s defense mechanisms during and after exercise. During exercise, the increased production of ROS and the imbalance in ATP/AMP ratios stimulate the activation of intracellular antioxidant systems to counteract oxidative damage. Interestingly, the recovery period following exercise presents a distinct phase characterized by a shift in antioxidant frontlines. Evidence suggests that during this period, certain antioxidants, such as polyphenols and flavonoids, may exhibit heightened activity. Polyphenols and flavonoids, which are found abundantly in the three natural extracts, are known for their potent antioxidant properties. They can scavenge free radicals and modulate oxidative stress pathways, contributing to the restoration of cellular homeostasis during recovery. Moreover, rapid-acting antioxidants such as glutathione (GSH) and ascorbic acid (vitamin C) play a crucial role during or immediately after exercise. These antioxidants act promptly to combat the oxidative stress induced by physical exertion, aiding in the recovery process. Recognizing the temporal activation of different antioxidant frontlines during and after exercise is essential for understanding the comprehensive antioxidant response. Interestingly, polyphenols/flavonoids have lower uptake rates by gastrointestinal tract than other antioxidants when orally administered, which keeps them at very low concentration in plasma [53]. According to some estimates, only 5–10% of ingested polyphenols can be absorbed in the small intestine. Therefore, the effective doses of these compounds in muscles are also very limited, which will clearly affect their mode of action as antioxidants. Rather than simple scavengers of ROS/RNS, it is possible that these compounds act as main activators of other antioxidant-responsive cascades in cells, e.g., the Keap1-Nrf2-AREs and NF-kB pathways [54,55]. It cannot be excluded that the Powergrape^®^ and/or other natural extracts tested in the present study activate these signaling cascades after acute flavonol/polyphenol supplementation. Further research is warranted.

Our study acknowledges some limitations. The use of an animal model may have some limits of transferability to humans, such as the lack of subjective assessment. The animal model was the only way to assess the function of the muscular tissues independently of cardiac load conditions and/or circulating factors. We did not evaluate the performance per se in this study; rather, we investigated the consequences of an intense prolonged exercise just after exercise or during recovery. One practical consequence is that for a similar performance, less cardiac alterations after exercise and better muscular recovery are observed if using the plant-derived antioxidants before. The administration and the doses of the extracts were consistent with human applications. The present study is a pre-clinical study aimed to identify the most efficient compound that could be tested with humans in a controlled experimental setting, which has been seen previously with Powergrape^®^ [12].

Another limitation is the lack of the biomarkers of metabolic and endocrine indexes of fatigue and oxidative stress to assess metabolic and endocrine responses to exercise. They provide valuable insights into the physiological changes occurring during intense physical activity and during the treatments. Under strenuous exercise, the body produces several hormones to maintain the energy metabolism [56]. Several biomarkers of energy metabolism and fatigue could have been investigated, including cortisol, glycogen, and lactate. Cortisol, a stress hormone, is commonly measured to evaluate the body’s response to physical stress. During intense exercise, cortisol levels typically increase, reflecting the activation of the hypothalamic–pituitary–adrenal axis. Measuring cortisol levels before and after exercise can provide insights into the stress response and recovery patterns, as well as the potential for overtraining or inadequate recovery. It is helpful in managing inflammation levels, sodium and water retention levels, and blood sugar levels. Lactate levels serve as an indicator of anaerobic metabolism and can provide information about the body’s energy production and lactate clearance capacity. Monitoring lactate levels before and after exercise can help quantify the extent of metabolic stress [57]. Biomarkers of oxidative stress, such as reduced and oxidized glutathione levels, carbonylated proteins, or F2-isoprostanes, are important indicators of the body’s antioxidant defense system and oxidative damage. Monitoring these biomarkers during exercise would have allowed for the assessment of the balance between oxidative stress and the body’s ability to counteract it, particularly in the context of natural extract treatments. This information is important for understanding the impact of exercise on cellular function and overall health. It is worth noting that the absence of these measurements does not invalidate the study’s findings; however, it does highlight an opportunity for future research to explore these metabolic and endocrine indexes to gain a more comprehensive understanding of the effects of exercise and potential treatments.

## 5. Conclusions

Our study shows that natural antioxidant plant extracts can protect the muscles during prolonged, exhausting exercise. Based on their impact on oxidation, some extracts have beneficial protective effects on both cardiac and skeletal muscles in the post-exercise period. These effects are even more pronounced during the one-week recovery period. However, whether these protective effects are applicable to professional athletes after strenuous or exhaustive exercise remains to be investigated.

## Figures and Tables

**Figure 1 antioxidants-12-01474-f001:**
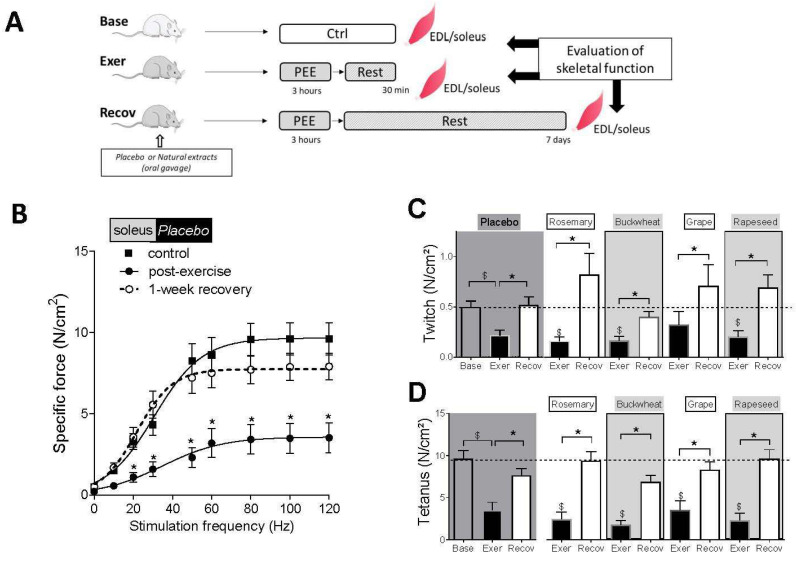
Effects of the 4 different natural extracts on the isometric strength of soleus after exhausting exercise and one week of recovery. (**A**) Soleus muscles were isolated from Sedentary (Base), 30 min post-exercise (Exer), and after one-week recovery (Recov). (**B**) Force–frequency relationships of soleus muscles measured in vitro from Sedentary (black square), 30 min post-exercise (black circle), and after one week of recovery (open circle, dash line) placebo groups. Results are expressed as mean ± SEM, (*n* = 10–13 animals/group). * *p* < 0.05 Placebo post-exercise vs. Basal. (**C**,**D**) reveal the different contractile characteristics measured in vitro of the Soleus muscles in groups of animals fed with the 4 different natural extracts (rosemary, buckwheat, Powergrape, and rapeseed), either immediately post-exercise or one week later. (**C**) represents the force induced by a single twitch, and (**D**) represents the tetanic contraction or peak specific force. Results are expressed as mean ± SEM, (*n* = 7–13 in each group). $ *p* < 0.05 vs. basal, * *p* < 0.05 recovery vs. post-exercise.

**Figure 2 antioxidants-12-01474-f002:**
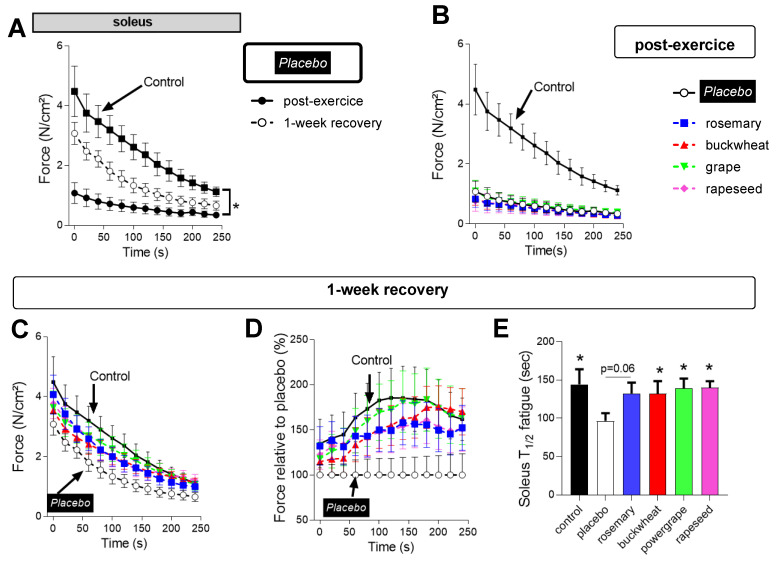
Effects of the 4 different natural extracts on the soleus fatigability after exercise and recovery. (**A**) Test of fatigue in vitro on soleus muscles in sedentary rats (control, black square), placebo post-exercise (black circle), and placebo one-week recovery (open circle) groups. (**B**) Effects of rosemary (blue square), buckwheat (red triangle), Powergrape^®^ (green triangle), and rapeseed (pink diamond) on in vitro post-exercise muscle fatigue test in soleus compared with placebo (open circle) and controls (black square). (**C**–**E**) Effects of the natural extracts on the in vitro muscle fatigue test one week after exercise compared with the placebo (open circle) and controls (black square). (**D**) Muscle fatigue was also expressed relative to the level of force of the placebo group at each time point. (**E**) Time to 1/2 loss of maximum strength during fatigue test in the different groups after one week of recovery. (*n* = 7–13 animals per group). * *p* < 0.05 placebo vs. Control, rosemary, buckwheat, Powergrape^®^, and rapeseed groups.

**Figure 3 antioxidants-12-01474-f003:**
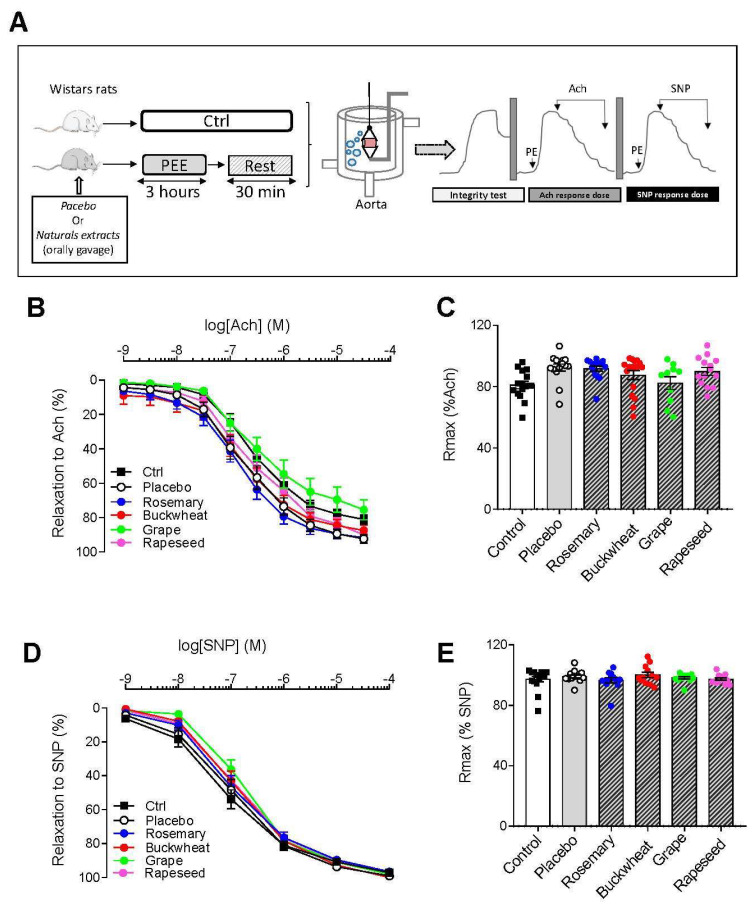
Effect of natural extracts on aortic endothelium-dependent and endothelium-independent vasorelaxation after prolonged exhausting exercise. (**A**) Schematic representation of the experimental design. Dose–response curves (**B**) and maximal response (**C**) to acetylcholine (ACh) obtained in aortic rings pre-contracted with phenylephrine (PE, 10^−6^ M) (*n* = 13–15 aortic rings from 5–6 different rats per condition) for control (Ctrl, black square), placebo (open circle), Rosemary (blue circle), Buckwheat (red circle), grape (green circle), and rapeseed (pink circle). Each dot is one preparation. Dose–response curves (**D**) and maximal response (**E**) to sodium nitroprusside (SNO) obtained in aortic rings pre-contracted with phenylephrine (PE, 10^−6^ M) (*n* = 13–15 aortic rings from 5–6 different rats per condition). All values are the mean ± SEM.

**Figure 4 antioxidants-12-01474-f004:**
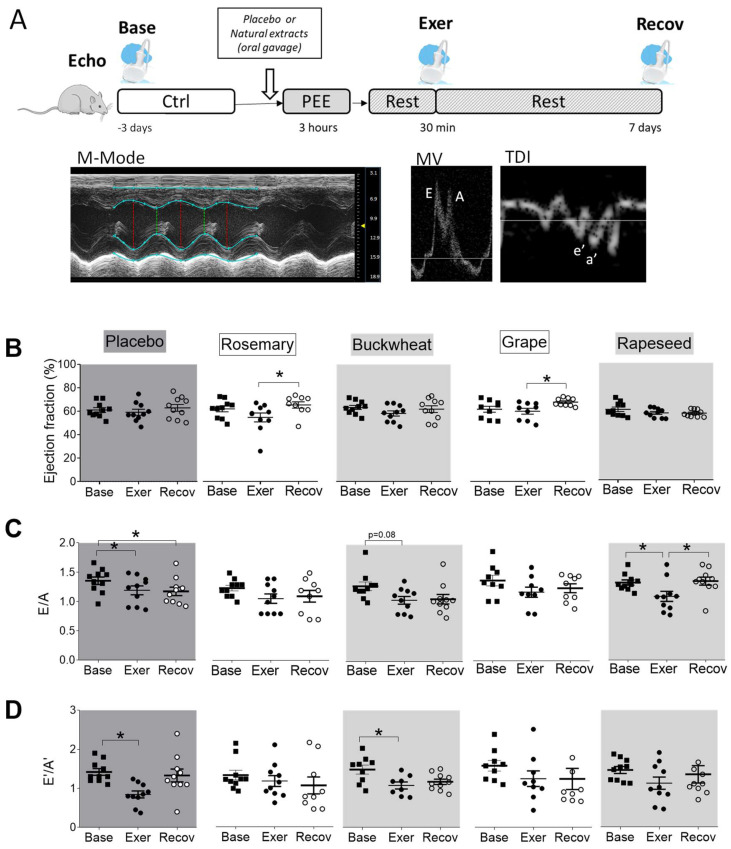
Effects of natural extracts on in vivo cardiac function after exercise. (**A**) The cardiac function was evaluated in vivo by echocardiography at baseline (base, black square), 30 min after PEE (Exer, black circle), and 1 week after exercise (Recov, open circle) in placebo and extract-treated animals. (**B**) The function of contraction or systole was indexed by the ejection fraction. The phase of relaxation or diastole of the heart was indexed by the E/A ratios of early diastolic and atrial transmitral inflow velocities (**C**), as well as the E′/A′ ratios obtained by tissue Doppler imaging (**D**). (*n* = 10/group). * *p* < 0.05 repeated measures ANOVA.

**Figure 5 antioxidants-12-01474-f005:**
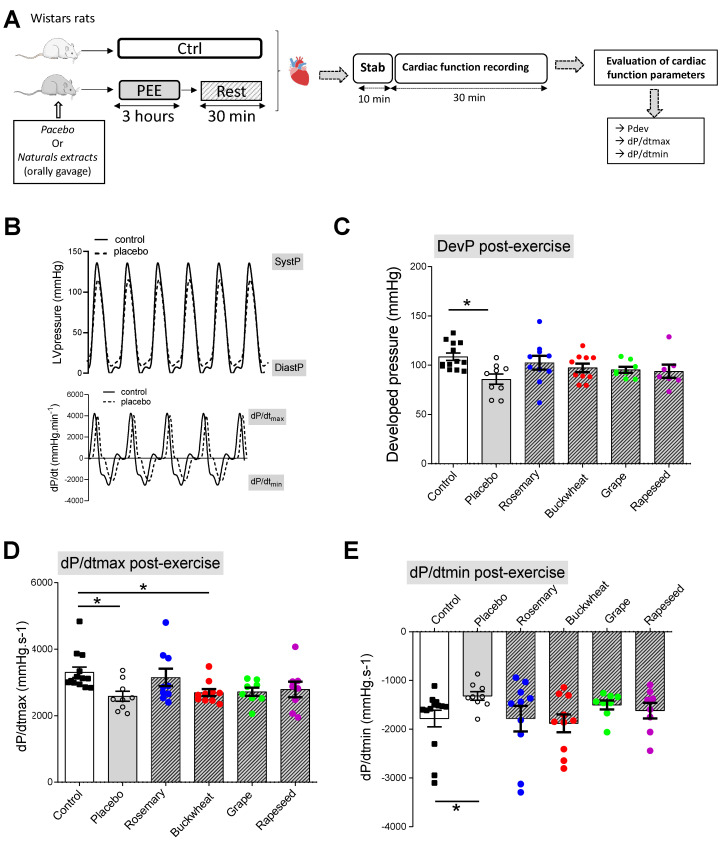
Effects of natural extracts on ex vivo cardiac function after exhausting exercise. (**A**) Schematic representation of the experimental design. (**B**) Representative trace of the left ventricular (LV) pressure (upper panel) and its first derivative (lower panel) obtained from Langendorff hearts isolated from Ctrl and Placebo-exercised rats. (**C**) LV developed pressure obtained from hearts 30 min after exhausting exercise in rats treated or not treated with 4 different natural extracts (*n* = 7–13 hearts per condition). Maximal first derivative of LV pressure (dP/dtmax) (**D**) and minimal first derivative of LV pressure (dP/dtmin) (**E**) obtained from Langendorff isolated hearts 30 min after exhausting exercise in rats treated or not treated with 4 different natural extracts (*n* = 7–13 hearts per condition). * *p* < 0.05 vs. Ctrl ANOVA followed by adjusted *t*-test; All values are the mean ± SEM.

**Figure 6 antioxidants-12-01474-f006:**
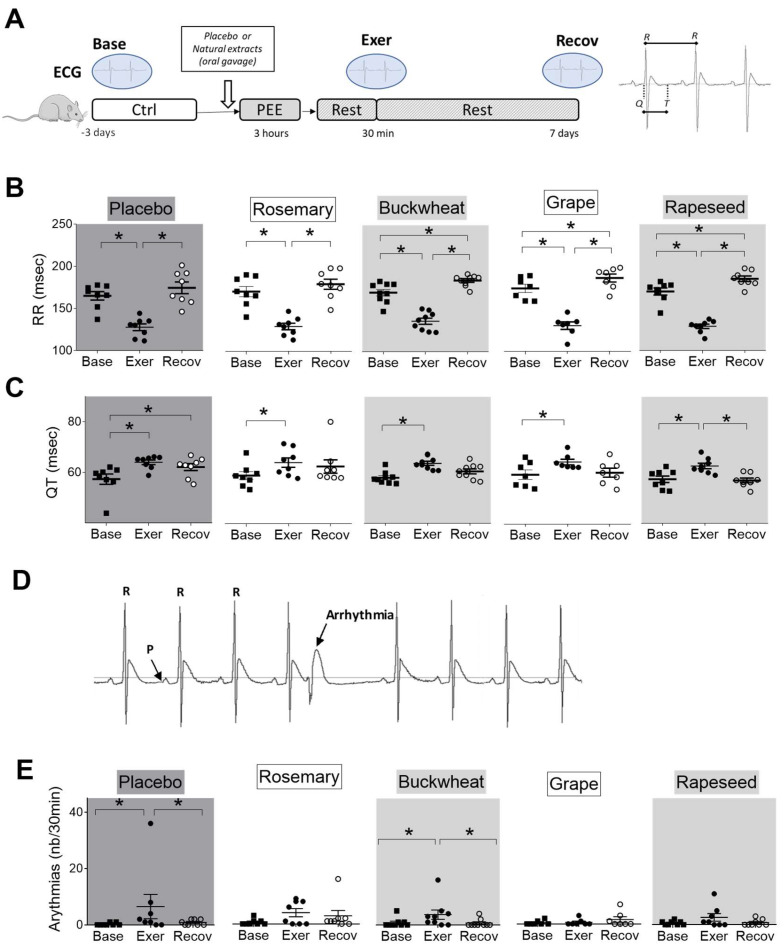
Effects of natural extracts on in vivo electrical cardiac function after exercise. (**A**) The ECG was recorded in vivo for the same rat at baseline (base, black square), 30 min after PEE (Exer, black circle), and 1 week after exercise (Recov, open circle) in placebo and extract-treated animals. (**B**) Heart rate (RR interval in ms) and (**C**) QT duration (in ms) were analyzed. (**D**) Example of ECG recording of a rat with ventricular extrasystole. (**E**) Number of ventricular extrasystoles (number/30 min) in extract-treated animals. (*n* = 8/group). * *p* < 0.05 repeated measures ANOVA.

**Table 1 antioxidants-12-01474-t001:** Summary of the effects of intense exercise on cardiac and muscular functions in presence or absence of antioxidant natural extracts.

	Post-Exercise					Recovery				
	Placebo	Rosemary	Buckwheat	Powergrape	Rapeseed	placebo	Rosemary	Buckwheat	Powergrape	Rapeseed
**Cardiac function**										
Contraction	**=**	**=**	**=**	**=**	**=**	**=**	**↗**	**=**	**↗**	**=**
Relaxation	**↘**	**=**	**↘**	**=**	**↘**	**↘**	**=**	**↘**	**=**	**=**
QT duration	**↗**	**↗**	**↗**	**↗**	**↗**	**↗**	**=**	**=**	**=**	**=**
Arrhythmias	**↗**	**=**	**↗**	**=**	**=**	**=**	**=**	**=**	**=**	**=**
**Muscular function**										
EDL_FF	**↘**	**=**	**↘**	**=**	**=**	**=**	**=**	**=**	**=**	**=**
EDL_fatigue	**↘**	**↘**	**↘**	**↘**	**↘**	**↘**	**=**	**=**	**=**	**=**
Soleus_FF	**↘↘**	**↘↘**	**↘↘**	**↘↘**	**↘↘**	**=**	**=**	**=**	**=**	**=**
Soleus_fatigue	**↘↘**	**↘↘**	**↘↘**	**↘↘**	**↘↘**	**↘**	**=**	**=**	**=**	**=**
Cardiac Protective effect Muscular Protective effect		++ +	0 0	++ +	+ +		**+** **+** **+** **+**	**+** **+** **+**	**+** **+** **+** **+**	**+** **+** **+** **+**

= no change between post-exercise and baseline control; ↗ increase or ↘ decrease post-exercise when compared with baseline control (the double arrow means large effect). 0: no cardioprotective effect, +: cardioprotective effect, ++: important cardioprotective effect.

## Data Availability

Full data are available from the corresponding authors upon reasonable request.

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
