# Peer review of "Natural Extracts Mitigate the Deleterious Effects of Prolonged Intense Physical Exercise on the Cardiovascular and Muscular Systems"

_antioxidants, 2023, doi:10.3390/antiox12071474_

Round 1

Reviewer 1 Report

This is an interesting animal study, well designed and executed. I have several comments, which might improve the manuscript:

1. Introduction is too long, parts of it should be moved to the discussion section. It should not be longer than 1-1,5 pages and present only the essentials to understand the purpose of the study. 

2. Why have you chosen to study animals and not athletes? Please discuss the limitations related to this model and its transferability to humans such as lack of subjective assessment? 

3. Have you performed power and sample size calculations? There are many comparisons made and therefore some of the findings might have been related to false positive results. Please discuss.

4. What might be the practical consequences of the results - does lower fatigue transfers to better physical performance in the acute or prolonged phase? This was not assessed, please discuss.

5. The main issue from cardiological point of view is the potential risk of pro-arrhythmic effect of various natural substances, especially when they are mixed? Could you comment on that. 

None

Author Response

First of all, we would like to thank you for taking the time to consider our manuscript and to suggest some relevant improvements. You should find below our replies to your comments and recommendations, as well as each modification within the manuscript (word tracking). We hope this revised version will meet the requirements of your journal for publication.

Reviewer 2 Report

I thank the authors for allowing me to review their manuscript. The manuscript is potentially interesting and highlights the properties that a supplementation with these natural extracts has in the prevention/contrast of problems related to physical exercise. However there are some critical issues that do not allow publication in the journal Antioxidants in this version.

Among these the major ones are:

1)      The authors should indicate the percentage of molecules most present (indicating the quantity) in the extracts. At least those with known bioactive role indicating the percentage.

2)      Then they should comment on the possible mechanism of action of these molecules in the various tissues also on the basis of what is known in the literature.

3)      Authors should update all literature (there are very old references) by updating to the latest news on the subject.

Author Response

(The authors gave the same response as above.)

Reviewer 3 Report

Yehya et al. studied here the protective properties of four renowned antioxidant plant extracts on the contractile funstions of skeletal and cardiac muscles of Wistar rats during/after exhaustive running exercises.

The MS is well-written and it is relatively clear on its aims. The subject is not new as many polyphenol/flavonoid-rich extracts have been tested against fatigue effects based on their antioxidant properties. In fact, there is certain urge for the proper mechanisms by which those compounds actually promote their beneficial effects on performance and/or recovery and health.

The study focused on key and accurate physiological parameters to  monitor skeletal and cardiac contractile activities, but important metabolic and endocrine indexes were  missing: the authors should have measured lactate and cortisol levels (pre/post effort, aiming to quantify it), as well as, key biomarkers of oxidative stress, such as reduced and oxidized glutathione levels in plasma at those specific checkpoints. If it is impossible to measure that, I presume one robust paragraph, at least, should mention and discuss that.

 Unbalances in ATP/AMP ratios in muscles are known to activate the purine catabolic pathway, which culminates in uric acid production in muscles and vascular system. Paradoxically, uric acid is a powerful antioxidant, per se, whereas xanthine oxidase, an enzyme of the purine pathway, produces reactive oxygen species during its catalytic steps. I think it is important to comment that different antioxidant frontlines are activated during physical effort and after it. Maybe polyphenols/flavonoids are more active during the recovery period, since GSH and ascorbic acid are apparently the promptly responsive ones. Again, at least one paragraph in discussion seems necessary to address that.

Finally, polyphenols/flavonoids have lower uptake rates by GI tract (orally administered) than other antioxidants, which keeps them at very low concentration in plasma. Therefore, the effective doses of these compounds in muscles are also very limited, which will clearly affect their mode of action, as antioxidants: rather than simple scavengers of ROS/RNS, these compounds are now accepted as main activators of antioxidant-responsive cascades in cells, e.g. the Keap1-Nrf2-AREs and NF-kB pathways!  So, the authors have to discuss their results based on these references, and, obviously, on the activation of these signaling cascades after acute flavonol/polyphenol supplementation. 

Minor English grammar/style issues along the text. Please revise MS once again for them.

Author Response

(The authors gave the same response as above.)

Round 2

Reviewer 2 Report

The authors responded sufficiently to the critical points raised. The manuscript can be accepted for publication.

Reviewer 3 Report

I am glad the authors understood my major concerns on their study and properly improved the quality of discussion, by also adding the worthy mentioning limitations of it. THis will profoundly help investigators worldwide for further studies. Thank you for the opportunity to help.